# A source data privacy framework for synthetic clinical trial data

**Afrah Shafquat**
Medidata Solutions, a Dassault Systèmes company
New York, NY 10014
Afrah.Shafquat@3ds.com

**Jason G. Mezey**
Department of Computational Biology, Cornell University
Ithaca, NY 14850
Department of Genetic Medicine, Weill Cornell Medicine
New York, NY 10065
jgm45@cornell.edu

**Mandis Beigi**
Medidata Solutions, a Dassault Systèmes company
New York, NY 10014
Mandis.Beigi@3ds.com

**Jimeng Sun**
Computer Science Department, University of Illinois Urbana-Champaign
Champaign, IL 61820
jimeng@illinois.edu

**Jacob W. Aptekar**
Medidata Solutions, a Dassault Systèmes company
New York, NY 10014
Jacob.Aptekar@3ds.com

## Abstract

Synthetic clinical trial data create opportunities for data sharing, cross-collaboration, and innovation for these valuable, siloed data sources. While the value of synthetic clinical trial data relies on the privacy preservation it offers the clinical trial participants, the true degree of privacy has been questioned in recent literature. Given the highly sensitive nature of clinical trial data, especially their content composing private health information, there is an urgent need for a framework specifically designed to provide guaranteed levels of privacy for synthetic datasets generated from clinical trial data. In this paper, we propose a practical privacy framework that ensures synthetic clinical trial data privacy at the level of the source data by design and provides objective, measurable bounds on the disclosure risks through a combination of technical, policy, and algorithmic controls. The proposed framework enforces privacy prior to the generation of synthetic datasets and therefore complements the privacy preserving attributes intrinsic to the algorithms used for synthetic data generation. To demonstrate how the components of the framework address the privacy requirements needed for clinical trial data, we discuss how this privacy system responds to a set of realistic adversarial scenarios.

NeurIPS 2022 Workshop on Synthetic Data for Empowering ML Research.

Ultimately, we believe the proposed framework can foster more privacy research in clinical trial data sharing.

# 1 Terminology

For the benefit of readers not familiar with clinical trials, we provide the following definitions:

- **Patient**: Participant in a clinical trial
- **Clinical trial or study**: Research study used to evaluate a new medical, surgical, or behavioral intervention
- **Clinical trial sponsor**: Entity or Organization monetarily responsible for conducting the clinical trial
- **Data contributor**: Entity or Organization whose data is included in training a synthetic data generation model
- **Data broker**: Entity or organization responsible for synthetic data generation
- **Data consumer**: Entity or organization that acquires synthetic data generated by data broker
- **Clinical data management organization**: Entity or organization responsible for capturing and managing clinical trial data using electronic data capture tools

# 2 Introduction

Synthetic data generation, the production of realistic data from a real data source, has shown promise in boosting performance of machine learning models by improving the quality [1–3] and quantity of training samples while preserving the privacy of the individuals. In the context of clinical trials, the ability to share synthetic clinical trial data while preserving patient privacy has been touted as a strategy for improving drug safety, evaluating bias, and other meta-analysis of multiple clinical trial studies [4, 5]. However, recent research has questioned the privacy protection provided by synthetic datasets [6–8], where in the context of health-related data, preserving privacy is critically important given moral obligations and detriment to patients if disclosure were to occur [9–11]. Furthermore, penalties of HIPAA violations for the organization responsible for the protection of such data can be in the tens of millions of dollars [12, 13]. Distinct from other industries, the presence of protected health information requires a conservative approach when sharing data, even synthetic data, in the healthcare domain.

For clinical trial datasets, it is imperative that synthetic data generation adapts a secure privacy-preserving framework to protect the privacy of patients in the source clinical trial datasets that are used to train the synthesis models. Several synthetic data generation methods guarantee individual-level privacy through privacy-preserving mechanisms such as $k$-anonymity, and $l$-diversity [14–17], however recent research has shown vulnerabilities in these approaches [18] highlighting the need for a different approach that can protect privacy in case of an attack. Differential privacy, a widely researched approach for privacy protection in synthetic data generation [19], guarantees quantified levels of privacy in synthetic datasets. In the absence of governance and enforced privacy budgets for differential privacy, there are concerns regarding the privacy embedded in the generated synthetic datasets [19, 20]. Given the risk associated with clinical trial data synthesis, a privacy framework customized to address the challenges of clinical trial data privacy is needed.

Analogous to email, where security is a composition of policy controls (e.g., password expiration, password length, complexity, prohibition of password sharing), technical controls (e.g., automatic logout, access only while on VPN, multi-factor authentication), and algorithmic controls (e.g., encryption-in-transit, encryption-at-rest), secure systems rely on layers of controls to achieve robust security. To ensure the privacy of clinical trial data, an open and effective privacy standard is needed such that it (i) protects patients from unwanted disclosures and financial or personal harm, (ii) allows institutions to contribute data by upholding their legal and ethical commitments to their patients, and (iii) supports the adoption of realistic synthetic data as an effective means of sharing useful information while protecting critical privacy interests (e.g., identities of clinical trial sponsors, clinical trial studies, and patients). In this paper, we propose a source data privacy framework that

increases the privacy protection upstream of synthetic data generation at the level of the source data. Consequently, the overall privacy of the generated synthetic dataset is inherited from the privacy delivered through this framework, enhancing privacy mechanisms intrinsic to the synthetic data generation model. Through this framework, the role of the synthetic data generation model transforms from being the main privacy delivery mechanism to being the enhancer of privacy for a dataset that has already been transformed to be strictly secure against contributor-level, study-level and individual-level identification.

## 3 A source data privacy framework for clinical trial data

To address the challenge of clinical trial data privacy, it is important to understand the provenance of clinical trial data. Clinical trials aim to capture information that can measure efficacy of a medical, surgical or behavioral intervention [21] while monitoring any related side effects, toxicity or adverse events caused by the intervention under investigation. To that end, clinical trial data typically contains patient medical history, laboratory test results, treatment administration schedule, medical visits, concomitant medications, drug response, and adverse events. Clinical trial patient population can be in the order of tens, hundreds or thousands of patients depending on the stage or phase of the clinical trial [21]. Patient data in clinical trials is collected using Electronic Data Capture (EDC) tools by a clinical data management organization. The workflow for clinical trial data generation follows: (i) capture of patient data (e.g. medical histories, adverse events, drug response) through EDC, (ii) quality controls and checks by clinical data management organization, and (iii) delivery of EDC data in Case Report Forms (CRFs) to clinical trial sponsor to generate reports and publication that support regulatory (e.g., FDA) approval of new interventions for disease. EDC data is accessible to the clinical trial sponsors and the clinical data management organization. The use of EDC by the clinical data management organization may be limited by contractual constraints as defined by the Data Use Agreement (DUA) with the clinical trial sponsor. When permitted by the DUA, the clinical data management organization may use the EDC data to provide insights to its customers, provided that the privacy of the data contributor's identity (i.e. sponsors of clinical trials used to train the synthetic data generation model) and clinical trial study are maintained, which requires safeguarding contributor-level, study-level, and individual-level privacy for clinical trial data.

The proposed privacy framework aims at increasing privacy using a series of technical, policy, and algorithmic controls upstream of the synthetic data generation process such that the source data ingested by the synthetic data generator protects the privacy of the contributor-level and study-level attributes even in its raw, unaltered form. The transformations are targeted toward improving the resulting dataset privacy and utility. We assume that EDC data is available to data contributors as case report forms (CRFs), where data contributors (and their respective customers) may only have access to CRFs for their own trials or those of their partners. Apart from a data leak within the data broker (entity responsible for synthetic data generation), the attacker may never access the data generated from the privacy framework's components. Figure 1 summarizes the proposed privacy framework where the components are (i) Minority of class, (ii) Compartmentalization, (iii) Federation, (iv) Obfuscation, (v) Standardization, (vi) Concatenation, (vii) Deletion, and (viii) Synthetic data generation. Each component of this framework and its privacy preservation mechanism are described below.

**Minority of a class** The cohort of clinical trials included to generate the synthetic data represents only a subset/minority of the total population of patients with the disease. Here the probability for a patient $p$ being in this cohort of clinical trials used to generate the synthetic data is given by (i) the probability that a clinical trial $t$ for disease $X$ was included in the synthetic data generation and (ii) the probability that a patient $p$ with disease $X$ was enrolled in clinical trial $t$. To note, this probability will be lower if $T$ (total number of clinical trials for disease $X$) or $P$ (total number of patients with disease $X$) is sufficiently large.

**Compartmentalization** The access policies for data extracts and any transformations within the privacy framework are enforced by the data broker such that only the data broker has access to the full lineage (e.g. clinical trials included, identification of data contributors), and set of transformations performed on the clinical trial datasets. This component asserts that no entity apart from the data broker responsible for synthetic data generation may have access or control to the source clinical trial datasets and any intermediate datasets produced by any of the following components. The separation

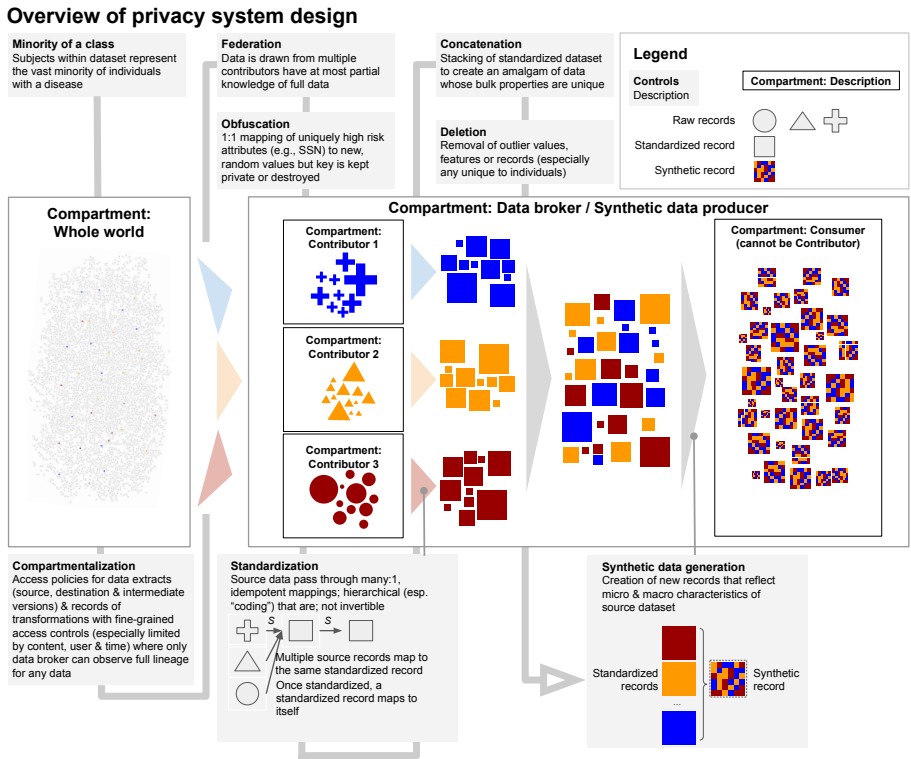

Figure 1: Overview of privacy system design for enhancing privacy of synthetic clinical trial data.

of access allows privacy of the identity of the data contributors and the clinical trials included to be preserved.

**Federation**    To preserve study-level and contributor-level clinical trial data privacy, synthetic datasets should be created from a randomized sampling of aggregated clinical trial datasets such that the final cohort of clinical trials used to train the synthetic data generator spans multiple clinical trial sponsors. This ensures that all resulting data contributors will always have partial knowledge of the final dataset used to train the synthesizer. This federation approach of access through aggregation and randomized sampling is limited to diseases where multiple clinical trial datasets have been conducted such that aggregation is possible. Inclusion of distinguished trials in terms of scale (or other characteristics) or limitations in aggregation may increase the contributor-level and study-level disclosure risk. To note, randomized sampling per clinical trial can increase ambiguity of a patient being included in the final cohort even if a clinical trial is disclosed to be part of the synthetic dataset, reducing the overall probability of the patient being in the dataset.

**Obfuscation**    To reduce disclosure risk, any high-risk identifiers as defined by the Safe Harbor method [22] should be mapped using one-to-one mapping to random codes (e.g. universally unique identifiers or UUIDs) where the key used to create these codes are either kept private or destroyed by the data broker. The obfuscation should be extended to any attributes that may be used for uniquely identifying patients, the clinical trial study (e.g. project identifiers, non-generic drug name, web links to project or clinical trial), clinical trial site location and/or data contributor-related information (e.g. clinical trial sponsor name, web links to clinical trial sponsor). For example, any geographical information provided should be mapped to a more generalized level such that the location indicated should have a sizable population. Obfuscation has also been referred as de-identification in other resources [22].

**Standardization**    To improve data quality, utility, and privacy, it is essential that all clinical trial datasets as defined by the cohort are standardized by (i) removing data entry errors and (ii) reducing the redundancies (e.g., same medication entered with different names across different trials) across

different datasets. The standardization should be performed using a many-to-one mapping that provides a reasonable level of specificity while ensuring generalization across the aggregated clinical trials such that specific values in the datasets may not be used for disclosing study and contributor identity. It is important to note this step is not invertible. Given the uniqueness in the strategy with which standardization is approached, a successful attack will require access to the final standardized dataset (only possible with a data leak) to be able to perform a one-to-one mapping to the exact categorical features and their standardized values. If the attacker only has access to the EDC/CRF data, the attacker will need expert domain knowledge to perform the many-to-one mappings and achieve the same level of standardization quality as achieved by the data broker during this step. Additionally, the probability the attacker performs the same many-to-one mapping and achieves the same final standardized dataset is also quite low.

**Concatenation**    This step allows the aggregation of standardized clinical trial datasets selected for synthesis. After concatenation, standardized datasets should have a uniform set of features. The resulting patient population from the aggregated standardized datasets should be homogeneous such that patients from different clinical trial studies should not be easily separable using unsupervised learning methods (e.g. clustering).

**Deletion**    To preserve the privacy of the overall dataset (i.e., patients, data contributors, studies), all outlier feature values or records should be removed from the dataset. Features values specific to a clinical trial such as specific treatment or non-generic drug names should also be removed. This step preserves $k$-anonymity of the dataset, which will prevent an attacker from using prior knowledge and elimination techniques to pinpoint certain individuals, sponsors and/or studies. Outliers may be identified using univariate (e.g. Interquartile range, confidence interval), multivariate (e.g. DBSCAN[23, 24], Isolation Forest[25, 26]) outlier detection techniques and model-specific methods (e.g. Cook's distance[27])[28].

**Synthetic data generation**    The last step of the process is synthetic data generation. Numerous techniques have been proposed for tabular data synthesis including Variational Autoencoders (VAEs) [29], LSTM models [30], CTGAN [31], Sequential trees [32], Patient Generator[33], EMRbots[33], Synthea[34], and other probabilistic network-based models. Temporal and sequential models have been proposed for longitudinal datasets such as Probabilistic Autoencoders (PAR)[35, 36]. Though the proposed privacy-preserving components listed above provide study-level and contributor-level privacy protection, to minimize individual-level disclosure risks associated with synthetic data generation [6–8], it is recommended to employ methods that incorporate differential privacy into their model training process such as PATE-GAN [37] and PrivBayes [38]. The proposed methods rely on adding noise to the objective function or the loss function in deep neural network models or in the case of marginal distribution-based methods to the distribution.

In addition to privacy enhancement through data-based controls provided by the proposed privacy framework, we recommend adopting economic and legal controls to mitigate disclosure risk. Contractually, data consumers should not be data contributors of the cohort, which would lower the risk of re-identification by data contributors. From an economic perspective, third-party users of the synthetic clinical trial data (i.e., not the data contributors studying their own data) should pay a monetary sum for access and should be required to provide similar access to their clinical trial data to contribute to the pool of data available to the data broker for creating synthetic data. Research has shown that costs can serve as deterrents to potential would-be adversaries and mitigate the chance that re-identification attacks will be perpetrated [39, 40]. From a legal perspective, there should be a contractual agreement that may be invoked to mitigate risk. Specifically, all consumers of synthetic data should enter into a contract that prohibits attempts at patient re-identification or inference of the data contributors' identity. For example, consumers of synthetic data should be prohibited from seeking clinical trial data directly from the data contributors. Similarly, consumers of synthetic data are prohibited from sharing this data with the data contributors.

## 4    Adversarial scenarios

The risk attributed to sharing clinical trial data and the financial liability in case of HIPAA violations [13] assert the importance for all synthetic clinical trial data (and for any other highly sensitive

data such as Electronic Health Records) to be evaluated for disclosure risk attached to contributor-level, study-level and individual-level re-identification. To understand the utility of this privacy framework, we consider how the system responds to different attack scenarios. We underline the key safeguards in the privacy system that protect against different adversaries possessing varying levels of information about the patients and the clinical trial data. The main disclosure risks in clinical trial datasets include (i) Membership disclosure risk, (ii) Contributor disclosure risk, and (iii) Attribute disclosure risk. Here, membership disclosure risk is defined as the likelihood the attacker correctly guesses that a subject is included in the training dataset used to generate the synthetic dataset [41, 6, 42, 43]. Contributor disclosure risk is the likelihood of re-identification of the data contributors. Attribute disclosure risk is the risk associated with an attack where the attacker tries to disclose sensitive attribute values for a specific individual. Considering 77% of all data breaches in 2015-2019 were in the healthcare sector [44] and the continued increase in the number of data breaches and costs associated[45, 12, 46], we consider all likely attack scenarios, attack adversaries, and the corresponding key disclosure risks. We summarize possible attacks as: (i) External attack, (ii) Contributor attack, and (iii) Omniscient attack. Figure 2 summarizes these adversarial scenarios where disclosure risk in each category (i.e., Membership, Contributor, and Attribute) is described. These examples demonstrate how the proposed privacy system exerts robust control over the most damaging and likely attack scenarios.

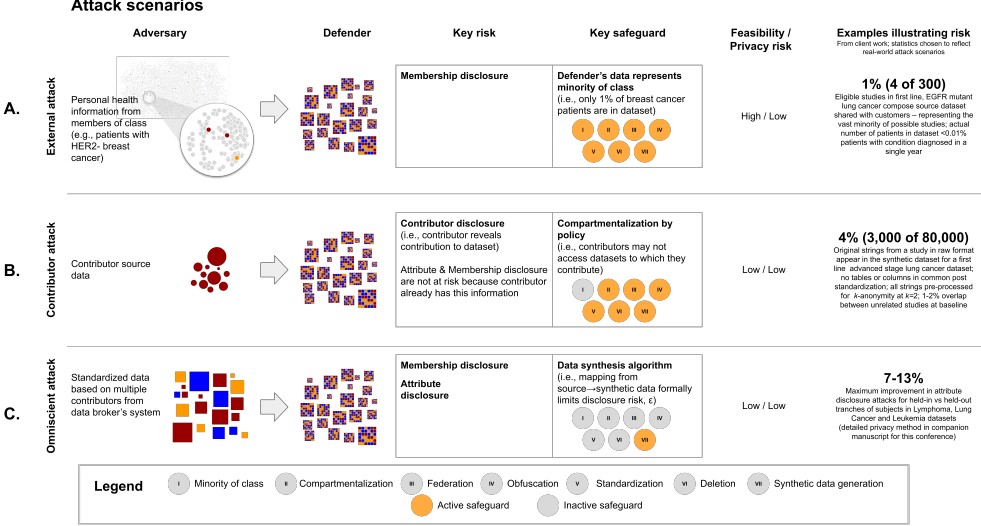

Figure 2: Attack scenarios for clinical trial data synthesis: (A) External attack, (B) Contributor attack, and (C) Omniscient attack. Adversaries with varying information about the clinical trial data are considered where the corresponding key risk, safeguard, and privacy risk is described for each attack scenario. Active safeguard describes framework components that actively protect against a specific attack and inactive safeguard describes framework components that don't protect against a specific attack.

**External attack**   External attack is a membership disclosure attack based on general knowledge about a class (Figure 2A). For example, a data consumer with access to a synthetic dataset on patients with pancreatic cancer wonders if an uncle who had pancreatic cancer is in the dataset. The most important safeguard is that the source data only represents a small minority of the potential data that could have been included; if 1% of individuals with pancreatic cancer are included in the dataset (baseline probability: 0.01), an attack would need to yield an odds ratio > 1,000 for the post-exposure probability of 0.95 that any particular individual were contained within the dataset [47]. In practice, only the presence of personally identifiable information (e.g., name, social security number) would allow such an attack to succeed.

**Contributor attack**   Contributor disclosure attack is perpetrated by a data contributor (Figure 2B). The risk of such an attack is minimized by policy if the contributor has agreed to share data and is also barred from accessing synthetic datasets to which they contribute. Even if the attacker gains

access to the synthetic dataset by breaching the data consumer's compartment, the most feasible attack is confirmation of their own contributions. Membership and attribute disclosure are moot as access to the source data from the contributor has already disclosed that information to the attacker (i.e., "why break the window if the door is unlocked?").

**Omniscient attack**    Omniscient attack is the well-defined source-to-synthetic data attack scenario that is typically described in publications on data synthesis techniques (Figure 2C). Within the proposed privacy framework, this scenario allows the data broker to quantify the upper bounds on the privacy risk and to measure the privacy protection provided by the data synthesis algorithm. However, the overall security of the system is a composition of these algorithmic safeguards–the degree of privacy delivered by the data synthesis algorithm itself–with policy, structural, and technical controls.

## 5   Conclusion

Sharing clinical trial data presents a unique challenge for preserving multiple layers of privacy i.e., on the individual-level, study-level, and contributor-level. Although the framework can accommodate other types of data, the proposed privacy system is specifically designed to address the challenges in clinical trial data synthesis. We also described possible adversarial scenarios to consider for evaluating protection provided by any privacy system. Generalized privacy mechanisms like differential privacy provide reasonable protection regardless of dataset type, however, additional controls are essential for datasets where disclosure can entail drastic monetary and discriminatory consequences for the disclosed entities (e.g. patients and clinical trial sponsors).

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
