# OpenReview forum: "A source data privacy framework for synthetic clinical trial data"
_NeurIPS.cc/2022/Workshop/SyntheticData4ML — Neurips 2022 SyntheticData4ML_

### Official Review · Reviewer_zzjs · 2022-10-05
**A source data privacy framework for synthetic clinical trial data**

**Rating:** 7
**Confidence:** 1

**Review:**

Review
1.	Quality
While the paper could be a useful contribution, and recognizing the fact that the submitted paper should be taken as a preliminary draft, subject to further development, the authors could consider how to better expand the contents of their proposed privacy enhancing framework. The authors start by introducing the requirements of a privacy enhancing framework by referring to
(i)	the protection of patients from unwanted disclosures and financial or personal harm;
(ii)	the allowance of institutions to contribute data by upholding their legal and ethical commitments to their patients, and
(iii)	the support of the adoption of realistic synthetic data as an effective means of sharing useful information while protecting critical privacy interests.
However, the paper does not offer a rhetorical discourse on how those requirements will be met and enhanced in relation to the present state of the art. Authors could consider structuring the paper differently to make their proposal more convincing.
2.	Clarity
The paper seems to be clear as to its objectives, the presentation of a privacy framework synthetic data use and sharing in clinical trials.
3.	Originality
The paper does not delve into its original character. It starts by briefly introducing some state-of-the-art privacy enhancing applications, but does not address whether, if at all, previous privacy enhancing frameworks similar to the proposed one have been already introduced. The paper goes then on to describe a standard clinical trial management system, but does not clearly address the way in which those processes will be enhanced by the proposed privacy framework. Next, the paper introduces the components of the proposed framework and, finally, attack scenarios. The conclusions seem to be underdeveloped and are not necessarily aligned with the discussions of the paper.
4.	Significance of this work, including a list of its pros and cons
While recognizing that this is an abstract, for the paper to be a useful contribution, it should introduce an analytical evaluation of the advantages of the proposed framework vis-à-vis the state of the art as well as the feasibility of the proposed framework, particularly in terms of cost and time. The conclusions could be better developed.

---

### Official Review · Reviewer_HUnz · 2022-10-17
**A source data privacy framework for synthetic clinical trial data**

**Rating:** 4
**Confidence:** 3

**Review:**

The paper provides a framework for achieving data privacy in clinical data before synthetic data generation. Data privacy is a very critical problem within clinical trials. Please see the feedback below:
1) The Paper is set up well but would be beneficial to add use cases of synthetic data generation to why that's needed. Although that's not the core contribution of the paper but improves readability.
2) In line 33-35, the paper claims privacy and synthetic data generation improves quality which is not true considering the noise introduced in the data which leads to a drop in the analytical utility of the data
3) Much synthetic data generation handles privacy during the generation part using differential privacy. What is the advantage of doing privacy before vs during synthetic data generation? Setting-up motivation for doing it before vs during synthetic data generation will help the reader.
4) One of the major concerns on the proposed framework is around compartments which also has led to the federated structure at the contributor level. The federation also brings data and model poisoning threats to the architecture which are not considered part of privacy.

---

### Official Review · Reviewer_YdKJ · 2022-10-19
**Innovative approach to data synthesis using sensitive data inputs**

**Rating:** 7
**Confidence:** 4

**Review:**

Well written and documented white-paper that proceeds a unique approach to better ensuring privacy and security of input data.

The downsides to this approach is that there is still risk associated with leveraging real world data as an input stream. This technique is more analogous to scrambling sensitive data rather than greenfielded data synthesis from statistical inputs or derived coefficients. A benefit to this approach is that in some cases, it will project out a very clear dataset that represents attributes present from the input stream. This would allow for a large variety of input streams to be used quickly. Downsides to this approach is that the data once generated, it is static. Having been synthesized or scrambled from real world data, it has no ability to further mutate or allow for historical lineage. For clinical trial usage, complex attributes will need to be retained such race, sexual orientation, geospatial attributes, ancestry, income, education etc.

I see powerful use cases for this approach, however care would need to be taken that the same type of data scrambling to produce the resulting dataset could not be used to reconstitute the original dataset using residual ontologies or taxonomies. This technique been exercised in high profile instances in the United States in recent years where sensitive input data was reconstituted from scrambled data causing significant invasions of privacy.

Good paper, certainly peaked my interest to learn more.

---

### Meta-Review · Area_Chair_72vi · 2022-10-20

**Recommendation:** Accept

**Review:**

Though the reviewers raise valid points for improvement, in general this seems to be outweighed by a good motivation, writing, and relevance to the community. I recommend to accept the paper.